

# The impact of short-term exposure to near shore stressors on the early life stages of the reef building coral *Montipora capitata*

Claire V.A. Lager[1,2], Mary Hagedorn[1,2], Kuʻulei S. Rodgers[2] and Paul L. Jokiel[2,†]

[1] Department of Reproductive Sciences, Smithsonian Conservation Biology Institute, Front Royal, VA, United States of America
[2] University of Hawaiʻi, Hawaiʻi Institute of Marine Biology, Kāneʻohe, Hawaiʻi, United States of America
[†] Deceased.

## ABSTRACT

Successful reproduction and survival are crucial to the continuation and resilience of corals globally. As reef waters warm due to climate change, episodic largescale tropical storms are becoming more frequent, drastically altering the near shore water quality for short periods of time. Therefore, it is critical that we understand the effects warming waters, fresh water input, and run-off have on sexual reproduction of coral. To better understand the effects of these near shore stressors on Hawaiian coral, laboratory experiments were conducted at the Institute of Marine Biology to determine the independent effects of suspended sediment concentrations ($100\,\mathrm{mg\,l^{-1}}$ and $200\,\mathrm{mg\,l^{-1}}$), lowered salinity (28‰), and elevated temperature (31 °C) on the successful fertilization, larval survival, and settlement of the scleractinian coral *Montipora capitata*. In the present study, early developmental stages of coral were exposed to one of three near shore stressors for a period of 24 h and the immediate (fertilization) and latent effects (larval survival and settlement) were observed and measured. Fertilization success and settlement were not affected by any of the treatments; however, larval survival was negatively affected by all of the treatments by 50% or greater ($p > 0.05$). These data show that early life stages of *M. capitata* may be impacted by near shore stressors associated with warming and more frequent storm events.

## INTRODUCTION

Coral reefs are among the most productive and diverse ecosystems in the world and these vulnerable ecosystems are rapidly experiencing global decline (*Bellwood, Folke & Nyström, 2004*; *Wilkinson, 2000*). They provide indispensable ecological services such as shoreline protection, food production, and are highly attractive to tourism (*Oliver, Lehrter & Fisher, 2011*). There are many stressors impacting the condition of coral reefs, both globally and locally. Global impacts include issues related to climate change such as sea surface temperature rise and ocean acidification (*McClanahan et al., 2007*; *Spalding & Brown, 2015*). Local impacts include fishing pressure, nutrification, coastal construction, dredging,

Corresponding author
Claire V.A. Lager, lagerc@hawaii.edu, lagerc@si.edu

increased sedimentation, invasive species, and freshwater runoff. All of the aforementioned stressors have been shown to negatively affect the condition of coral reefs (*Descombes et al., 2015*; *McManus et al., 2019*; *Banner, 1968*; *Fabricius, 2005*; *Hughes et al., 2007*; *Ogden & Lobel, 1978*; *Richmond, 1993*; *Rogers, 1990*).

Kāneʻohe Bay, (21°28′N; 157°48′W)—the largest embayment in the Hawaiian islands located on the windward side of Oʻahu—has been experiencing sedimentation and freshwater runoff for decades (*Bahr, Jokiel & Rodgers, 2015*). Freshwater runoff occurs during storm flooding and is a common event on tropical islands that can temporarily decrease salinity (*Banner, 1968*; *Jokiel et al., 1993*). Such events have been shown to cause mass mortality of adult coral (*Bahr, Jokiel & Rodgers, 2015*; *Banner, 1968*; *Jokiel et al., 1993*) and could pose a threat to the early life history of coral (*Babcock et al., 1986*; *Kolinski & Cox, 2003*). In the past 100 years, the corals of Kāneʻohe Bay have been chronically impacted by sediment, mainly through dredging and watershed runoff (*Bahr, Jokiel & Toonen, 2015*).

More recently, Kāneʻohe Bay has been experiencing warmer summer temperatures (2−4 °C above the summer average of 27 °C) that have resulted in coral bleaching. In 2014 and 2015, Kāneʻohe Bay experienced consecutive warming events that resulted in widespread bleaching of coral. With increased intensity and frequency of bleaching events coral are less likely to recover. *Bahr, Rodgers & Jokiel (2017)* surveyed coral before and after both the 2014 and 2015 bleaching events. They found that overall coral mortality was higher after the 2015 bleaching event (5.5% in 2014 and 16.0% in 2015) which suggests that consecutive bleaching events affect coral resilience.

The effects of local near shore stressors such as sedimentation, freshwater runoff, and elevated temperatures on adult coral have been thoroughly studied (*Erftemeijer et al., 2012*; *Fabricius, 2005*; *Humphrey et al., 2008*; *Jokiel et al., 2014*; *Rogers, 1990*; *Te, 2001*). More recently, studies have looked at the effects of near shore stressors on the early life stages of coral as well (*Edmunds, Gates & Gleason, 2001*; *Hedouin, Pilon & Puisay, 2015*; *Humanes et al., 2017*; *Jones, Ricardo & Negri, 2015*; *Ricardo, Clode & Humanes, 2015*; *Ricardo et al., 2018*; *Ricardo et al., 2017*). However, very little is known about how these near shore stressors affect the early life stages in Hawaiian coral. These near shore stressors were selected for this study because all three have been shown to negatively affect adult coral worldwide (*Douglas, 2003*; *Fabricius, 2005*; *Rogers, 1990*) and in Hawaiʻi (*Bahr, Jokiel & Rodgers, 2015*; *Jokiel & Brown, 2004*; *Jokiel et al., 1993*; *Jokiel et al., 2014*).

In this study we exposed gametes of the scleractinian coral *M. capitata* to three different near shore stressors for 24 h: high sediment concentrations (100 mg l$^{-1}$ and 200 mg l$^{-1}$), lowered salinity (28‰), and elevated temperature (31 °C) (Fig. 1). Exposure to stressors was independent and none were combined. Criteria selection for stressor levels include: (1) accurate representation of Kāneʻohe Bay conditions (Table 1), and (2) comparability with historical studies conducted with other Pacific species.

The purpose of this study was to determine whether the different stressors have deleterious effects on the early life stages of *M. capitata*: (1) fertilization, (2) larval survival, and (3) settlement. Successful development of early life stages of coral are very important for reef resilience and recovery. Therefore, understanding how these common stressors affect the early life stages is imperative.

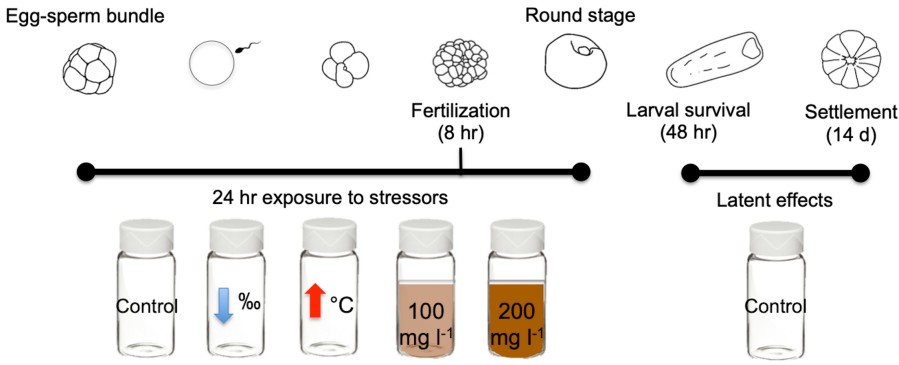

**Figure 1  Diagram of early life stages and exposure timing.** Diagram illustrating the 24 h exposure of stressors to the early stages of *M. capitata*. Exposure began prior to egg-sperm bundle breakup and lasted through the non-motile round stage. Fertilization success was measured at 8 h and 24 h. Then at 24 h, the embryos were transferred to clean vials (labeled treatment stressors) but now with filtered seawater with ambient salinity and temperature. These same treatments were assessed for larval survival at 48 h and were finally placed into petri dishes for settlement. Settlement was counted 14 days after spawning. Figure adapted from *Jones, Ricardo & Negri (2015)*.

**Table 1  Summary of normal and extreme water quality for Kāneʻohe Bay, Oʻahu.** This table summarizes the average/normal values and extreme (i.e., post-storm or bleaching) for suspended sediment (mg l⁻¹), salinity (‰) and temperature (°C) in Kāneʻohe Bay, Oʻahu.

|  | Normal | Extreme (i.e., storm or bleaching event) |
|---|---|---|
| Sediment | $13.1 \pm 0.45$ mg l⁻¹ (*Uchino, 2004*) | 600–800 mg l⁻¹ (*Hoover & Mackenzie, 2009*) |
|  | 1–3 NTU (*De Carlo et al., 2007*) | 8 NTU (*De Carlo et al., 2007*) |
| Salinity | 34–35‰ (*De Carlo et al., 2007*) | >20‰ (*De Carlo et al., 2007*) |
| Temperature | 27 °C—summer avg (*Bahr, Jokiel & Rodgers, 2015*) | 30–31 °C (*Bahr, Jokiel & Rodgers, 2015*) |

# METHODS AND MATERIALS

## Location and study species

These experiments were conducted at the Institute of Marine Biology (HIMB) in Kāneʻohe Bay (21°28′N; 157°48′W) on the windward side of Oʻahu, under the Hawaiʻi Department of Land and Natural Resources Special Activity Permit No. SAP 2015-48. *M. capitata* is the second most common coral species in Kāneʻohe Bay and the third most common throughout the Hawaiian Islands (*Jokiel et al., 2004*; *Rodgers, 2005*). *M. capitata* is a hermaphroditic broadcast-spawner, releasing positively buoyant egg and sperm bundles during the months of May through August between the hours of 22:45 and 22:30 on and 2–3 days following the new moon (*Kolinski & Cox, 2003*). Thirty-four adult coral were collected from multiple patch reefs throughout Kāneʻohe Bay on 14 May and 12 June, 2015 and transported back to tanks at HIMB. Coral were held in seawater tables continuously supplied with 27 °C seawater pumped from the adjacent reef at 2 m depth. Approximately one hour prior to spawning, coral colonies were isolated into individual containers. The exposure experiment was replicated on three spawning nights during July 2015 with a total starting sample size of approximately 33 for each treatment (Table 2). Vials with less than
**Table 2 Summary of sample sizes for the treatments.** This table lists the sample sizes of the different treatments at the three early life stages: fertilization, larval survival, and settlement. There was an inherent loss of sample size throughout the experiment.

|  | Fertilization | Larval survival | Settlement[a] |
|---|---|---|---|
| Control | 33 | 28 | 22 |
| Medium suspended sediment | 28 | 24 | 5 |
| High suspended sediment | 30 | 30 | 15 |
| Low salinity | 31 | 29 | 1 |
| High temperature | 29 | 21 | 12 |

**Notes.**
[a] Due to low sample size, differences in settlement were not analyzed statistically.

10 eggs were not included for analysis due to the high likelihood that only one egg-sperm bundle was added.

## Sediment treatment

Three concentrations of sediment were used: 0 mg l$^{-1}$, 100 mg l$^{-1}$, and 200 mg l$^{-1}$. These concentrations were chosen to mimic suspended sediment concentrations during/after a largescale rain event (Table 1). Terrigenous red clay was collected from a historically undisturbed hillside at the highest elevation on Moku o Lo'e Island at HIMB. Sediment was sorted using a standard sieve to <63 $\mu$m. This sediment size was chosen to mimic natural suspended sediment grain size. This clay/silt fraction was added to a 1 L beaker of seawater, allowed to settle and the clear supernatant decanted to concentrate into a sediment slurry that could be more accurately measured into 100 and 200 mg l$^{-1}$. The remaining sediment slurry was used to make sediment solutions. Sediments were not allowed to desiccate completely, which can alter the chemical composition and also impede re-suspension (*Jokiel, 1986*). A wet weight to dry weight ratio was determined in order to obtain an accurate suspended sediment concentration. The wet slurry was weighed and dried several times to obtain an accurate wet to dry weight ratio. The day of the experiment, the sediment slurry was weighed to the nearest milligram on a Mettler Toledo XS403S scale (Columbus, OH) in a plastic weigh pan and added to 1 L of filtered seawater (FSW). This was repeated for each sediment concentration. All FSW was filtered through a Millipore Type GS 0.22 $\mu$m filter.

## Salinity treatment

Fertilization, larval survival, and settlement of *M. capitata* was assessed at ambient (34‰) and low (28‰) salinity. Treatment seawater was obtained from the seawater system at HIMB and salinity was measured prior to use. Salinity was measured in parts per thousand (‰) using a YSI Model 556 conductivity meter (Yellow Springs, OH). The 28‰ treatment was prepared using filtered Kāne'ohe Bay seawater diluted with filtered freshwater to obtain the desired salinity. The salinity of 28‰ was selected based on previous studies which is representative of salinities measured on nearshore reefs during flood events (*Hedouin, Pilon & Puisay, 2015*; *Humphrey et al., 2008*).

## Temperature treatment

Two temperature treatments were used in these experiments, ambient (27 °C during the summer months), and an elevated temperature of 31 °C. The elevated temperature represents 2 °C above the summer thermal maximum, a temperature that elicits the stress response of bleaching in adult coral over a short time period (*Jokiel & Coles, 1977*). The elevated temperature of 31 °C was chosen because during the 2014 bleaching event in Kāneʻohe Bay the maximum mid-day temperature was between 30 and 31 °C (*Bahr, Jokiel & Rodgers, 2015*).

All ambient, sediment, and salinity vials containing gametes were secured in floating foam racks and were placed in water tables to maintain temperature (27 °C) and simulate mild wave motion similar to field conditions. The elevated temperature treatment (31 °C) vials were placed in a heated water bath and were also secured in floating foam racks. Temperature was controlled with an aquarium heater and an Onset HOBO Pendant® measured the temperature over the time of exposure. Loggers have an accuracy of $\pm$ 0.21 °C from 0 to 50 °C ($\pm0.38$°F from 32° to 122°F). Additional laboratory calibrations were conducted at 0 and 35 °C to assure precision and account for any drift in calibrations. Mild agitation was achieved using aquarium power heads that gently circulated water in the bath.

## Fertilization

Coral were placed into individual 11 L containers prior to spawning and water was allowed to flow in and around the containers to maintain constant water temperature. Unlike other fertilization experiments (*Gilmour, 1999*; *Hedouin, Pilon & Puisay, 2015*), egg bundles were not pooled since *M. capitata* eggs contain a toxin that will effectively kill sperm within minutes if their membrane is even slightly damaged (*Hagedorn et al., 2015*). Pooling eggs requires more handling and increases the chance of damaging membranes and releasing the toxin. Therefore, fertilization was accomplished by egg-sperm bundle-bundle crosses from two individuals in 15 ml scintillation vials (*Maté et al., 1997*). A "bundle-bundle cross" consisted of one egg-sperm bundle from each of two parent colonies that were loaded into a scintillation vial with 4.9 ml of filtered seawater (FSW). The final volume in each scintillation vial was 5 ml after the bundles were added. To determine the level of self-fertilization, two egg-sperm bundles from the same individual were placed in a vial—no colonies used in this experiment self-fertilized. The average sperm concentration of an *M. capitata* egg-sperm bundle is approximately $5 \times 10^5$ cells/ml (*Hagedorn et al., 2016*). The bundle-bundle cross method has been shown to produce the optimal sperm concentration ($\sim1 \times 10^6$ cells/ml) for fertilization in *M. capitata* (*Maté et al., 1997*). Over the three nights of spawning, 33 unique crosses were obtained. Each cross was exposed for 24 h to different treatments: (1) ambient conditions; (2) medium suspended sediment; (3) high suspended sediment; (4) low salinity; and (5) high temperature (Table 3), and (6) self fertilization control (2 bundles from same colony). Each treatment was separate and exposed to each cross in a scintillation vial for 24 h (Fig. 1).

Bundles separated approximately 20 min after spawning and the number of eggs per vial was recorded using a Wild M5 dissecting microscope at 100x magnification. The

**Table 3  Summary of conditions for the treatments and control.** There were five treatments—medium suspended sediment, high suspended sediment, low salinity, and high temperature—and the control. This table lists the suspended sediment, salinity and temperature values for each treatment and control.

| | Suspended sediment (mg l$^{-1}$) | Salinity (‰) | Temperature (°C) |
|---|---|---|---|
| 1. Control | 0 | 34 | 27 |
| 2. Medium suspended sediment | 100 | 34 | 27 |
| 3. High suspended sediment | 200 | 34 | 27 |
| 4. Low salinity | 0 | 28 | 27 |
| 5. High temperature | 0 | 34 | 31 |

fertilization count was recorded the following morning, approximately eight hours after spawning.

## Larval survival

Twenty-four hours after spawning with continued exposure to the treatments, the embryos were moved to clean vials with 5 ml of FSW in each vial. The number of swimming larvae was determined 48 h following spawning. The percent larval survival was calculated by dividing the number of swimming larvae by the number of fertilized eggs.

## Settlement

After counting the number of swimming larvae in each scintillation vial, larvae were moved into 10 ml petri dishes. A 1 cm$^2$ chip of crustose coralline algae (CCA) and approximately 10 ml of FSW were added to each dish and were covered loosely with a lid. Every two days fresh FSW was added to petri dishes and each dish and CCA chip were checked for settlement during water changes. Fourteen days after spawning, settlement was determined by counting the number of settled larvae. *Negri et al. (2001)* define settlement as a planula that is still pear-shaped but has attached its aboral end to a hard substrate, whereas metamorphosis involves a morphological and physiological change (i.e., flattened and with septal mesentery radiating from mouth). Larvae were considered "settled" if they had metamorphosed. The percent settlement was calculated by dividing the number of metamorphosed coral by the number of swimming larvae.

## Statistical analyses

All count data for fertilization, larval survival, and settlement were counted and represented as percentages. These values were transformed using an arcsine square root transformation as recommended for proportional data. Data are graphically displayed as percent in figures to better visualize the proportions and all error bars are Standard Error of the Mean (SEM). Each treatment was individually compared to the control using a One-way ANOVA with Dunnett's Method using the statistical software program JMP Pro 12. $P$-values $\leq 0.05$ were considered statistically significant.
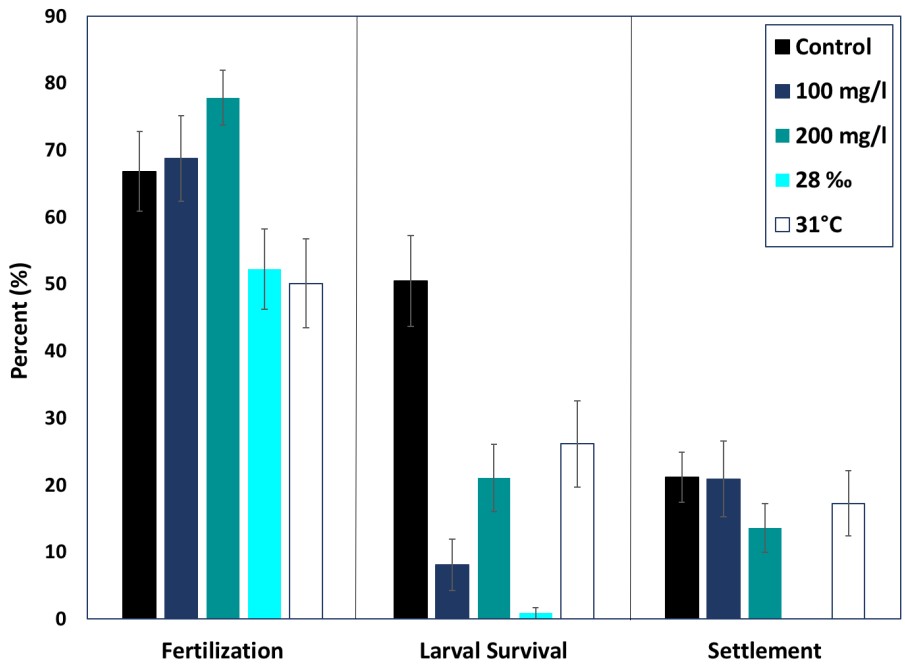

**Figure 2** **Effects of near shore stressors on early life stages.** The early life stages of *M. capitata* were exposed to increased suspended sediment, low salinity, and increased temperature. The percent fertilization, larval survival, and settlement for each treatment and control is graphed.

## RESULTS

### Fertilization

None of the treatments were statistically higher or lower than control fertilization success of *M. capitata* (ANOVA, $p > 0.05$). The mean fertilization success for ambient conditions was approximately $66.8 \pm 5.7\%$. All suspended sediment treatments showed slightly higher mean fertilization success; $77.8 \pm 4.05\%$ in the 200 mg l$^{-1}$ treatment and $68.7 \pm 6.3\%$ fertilization in the 100 mg l$^{-1}$ treatment (Fig. 2). The low salinity treatment ($28‰$, $n = 31$) decreased mean fertilization success to $52.2 \pm 5.9\%$, but was not statistically different from the ambient salinity of $34‰$ at $66.8 \pm 5.9\%$. The elevated temperature treatment (31 °C, $n = 29$) also produced a lower mean fertilization to $50.0 \pm 6.6\%$, but was not different than the control.

### Larval survival

Although fertilization appeared to be resilient to the treatments, larval survival (% of swimming larvae from fertilized eggs) decreased drastically in all of the treatments, especially hypo-osmotic conditions (Fig. 2). Control larvae under ambient conditions had the highest survival, $67.3 \pm 10.7\%$ (Fig. 2) and were statistically different than all of the treatments (ANOVA, $p < 0.05$). Embryos at the round, non-motile stage were removed from the treatments and placed into clean seawater for the rest of the experiment; differences in larval survival were due to latent effects of the treatments.

Both elevated sediment treatments had much lower larval survival when compared to the control ($p < 0.05$, $F = 13.9$). The 200 mg l$^{-1}$ sediment treatment had higher larval survival than the 100 mg l$^{-1}$ treatment ($24.47 \pm 5.5\%$, $11.6 \pm 5.2\%$). Elevated temperature produced the highest mean larval survival of all treatments, $36.4 \pm 8.6\%$. Salinity had the most dramatic effect on larval survival. The low salinity treatment had the lowest percent larval survival of all the treatments, $1.14 \pm 6.1\%$ (ANOVA, $p < 0.05$, $F = 38.8$).

### Settlement

There was no difference in percent settlement when treatments were compared with the control, $21.1 \pm 3.8\%$ (Fig. 2). However, the means were up to 50% different in the respective treatments. The 100 mg l$^{-1}$ sediment treatment had a greater mean percent settlement than the control ($39.5 \pm 14.3\%$), and the 200 mg l$^{-1}$ sediment treatment had a lower mean settlement ($20.0 \pm 6.1\%$) (Fig. 2). There was no settlement in the salinity treatment and, due to low survival, no statistical analysis was conducted. The high temperature treatment had a lower mean percent settlement than the control ($25.3 \pm 8.3\%$) (Fig. 2).

## DISCUSSION

The purpose of the present study was to identify the direct and latent effects of a 24-hour exposure to suspended sediment (100 mg l$^{-1}$ and 200 mg l$^{-1}$), lowered salinity (28‰), and elevated temperature (31 °C) on the early life stages of the Hawaiian scleractinian coral *M. capitata*. Results of this experiment show that fertilization and settlement were not affected by any of the stressors, and larval survival was negatively affected by all of the near shore stressors (suspended sediment, lowered salinity, and elevated temperature).

### Effects of increased sediment on early life stages

Neither suspended sediment treatment (100 and 200 mg l$^{-1}$) had an effect on fertilization but both decreased larval survival (Fig. 2). Previous and concurrent studies from the Great Barrier Reef found that sediment decreased both fertilization and larval survival (*Gilmour, 1999*; *Humphrey et al., 2008*). *Gilmour (1999)* found that suspended sediment as low as 50 mg l$^{-1}$ inhibited fertilization in *Acropora digitifera*. Additionally, *Humphrey et al. (2008)* saw reduced fertilization in *A. millepora* when exposed to suspended sediments (100 mg l$^{-1}$ and 200 mg l$^{-1}$). *Gilmour (1999)* exposed *Acropora digitifera* larvae to suspended sediment (20 mg l-1 and 50 mg l-1) and there was significantly greater mortality (>98%) in the sediment treated larvae.

*Ricardo, Clode & Humanes (2015)* found that fertilization decreased in the presence of suspended sediment, which was compounded by lowering the concentration of sperm available. When exposed to suspended sediment concentrations of 230 mg l$^{-1}$ and 700 mg l$^{-1}$, they determined that 2–37 fold more sperm was needed in order to equal fertilization rates seen in sediment-free treatments. In this study we used an optimal sperm concentration for fertilization in *M. capitata* (*Maté et al., 1997*) which may have made fertilization more resilient to the sediment treatments.

*Humanes et al. (2017)* performed similar experiments to this study but also looked at the combined effects of sediment, temperature, and nutrients on the early life stages of *Acropora*

*tenuis*. They found that fertilization was most sensitive to high suspended sediments (100 mg l$^{-1}$) while larval survival and settlement were not affected.

In , studies have shown that the coral in Kāneʻohe Bay show some resilience to sediment during gametogenesis and larval survival but not settlement. *Padilla-Gamiño et al. (2014)* looked at the effect of sediment on gametogenesis in *M. capitata* and found no difference in gamete production between sites with high and low sediment regimes. *Perez III et al. (2014)* exposed larvae of *Pocillopora damicornis* to substrate covered in varying levels of fine sediment (0.008–0.08 mm). They found that larval survival was not impacted by sediment but a thin layer (>0.9 mg cm$^{-2}$) of fine sediment could completely block settlement of larvae. *Ricardo et al. (2017)* also found that very thin layers of deposited sediment can block settlement and this was consistent regardless of sediment type (carbonate and siliciclastic) and particle size (fine and coarse silt).

Sediment type and composition has been shown to have varied effects on coral fertilization (*Ricardo et al., 2018*). Sediments with high organic-clay or certain minerals (i.e., Bentonite) decreased fertilization even at low suspended sediment concentrations. In contrast, terrigenous sediments with low organic matter only decreased fertilization at high suspended sediment levels (>100 mg l$^{-1}$). In this study we used terrigenous red clay but did not analyze the sediment for its organic or mineral composition. Low organic composition could explain fertilization resilience to suspended sediments seen in this study.

## Effects of low salinity on early life stages

Scleractinian corals are known to be stenohaline and osmoconformers. Corals do not have a developed physiological regulatory system thus, osmotic stress on corals may cause damage at the cellular level. A rapid increase in the induction of heat shock proteins may result from changes in salinity (*Seveso et al., 2013*). Lowered salinity did not have an effect on fertilization but did decrease larval survival and settlement. Kāneʻohe Bay has nine perennial streams that feed directly into it, which could explain why *M. capitata* appear to have some resilience to low salinity during fertilization. A few other studies have examined how low salinity affects fertilization, larval survival, and settlement but none have included Hawaiian coral. *Humphrey et al. (2008)* exposed *Acroporid* gametes to different salinities (28 to 36‰) and documented reduced fertilization at 30‰ and no fertilization at 28‰. Similarly, *Hedouin, Pilon & Puisay (2015)* exposed gametes of two *Acroporid* species to different levels (26 to 36‰) of salinity and found that salinities ≤ 28‰ (26.6 and 27.1‰) reduced fertilization success in both species. *Hedouin, Pilon & Puisay (2015)* also found that lowered salinity decreased larval survival. The results from the present study are more consistent with those of *Chui & Ang Jr (2015)*. They exposed gametes of *Platygyra acuta* to several different salinities and found that fertilization success was statistically the same from 32 to 28‰ with significant decreases at 26‰, suggesting that some species of coral may be more tolerant to lowered salinities.

## Effects of high temperature on early life stages

Exposing the early life stages of *M. capitata* to elevated temperature for 24 h following spawning had negative latent effects on larval survivorship but did not directly impact

fertilization. Gametes and embryos held at high temperature during this experiment had a lower mean fertilization but it was not different from the percent fertilization of the control. The results from the present study are consistent with studies performed on coral from Okinawa where *Negri, Marshall & Heyward (2007)* found that *Favites chinesis* had high fertilization success even at 31.8 °C (∼79–91%). Other studies have shown that elevated temperature negatively affects larval survival and settlement. *Bassim & Sammarco (2003)* and *Edmunds, Gates & Gleason (2001)* found elevated temperature treatments increased mortality in coral larvae. Coral settlement was also found to be negatively affected by elevated temperature (*Bassim & Sammarco, 2003*; *Randall & Szmant, 2009*). *Humanes et al. (2017)* found that elevated temperature (31 and 32 °C) decreased fertilization, larval development, and settlement in *Acropora tenuis*.

In this study, there was no statistical difference in the percent settlement of *M. capitata* larvae between the control and the treatments. However, the results of the settlement experiment are lacking due to low replication at the settlement stage (Table 2). Following the same cohort of gametes and embryos through fertilization, larval survival, and settlement resulted in important information about the lasting effects of a short-term exposure to stressors. However, this led to an inherent loss of sample size for settlement results.

## CONCLUSION

Kāneʻohe Bay is a calm, protected lagoon and *M. capitata* is one of the major reef building coral there. As climate change accelerates, more frequent episodic largescale storms and hurricanes will impact the main Hawaiian Islands (*Li et al., 2018*). Storms may produce a lens of warm hyposaline water with increased suspended sediments that result in high mortality in the early life stages of *M. capitata* when spawning is synchronous to flooding. Our study showed that these stressors did not affect fertilization, but there were negative latent effects on larval survival. Most notably, gametes and embryos exposed to salinity of 28‰ for 24 h had less than 1% larval survival while those exposed to sediment decreased larval survival by approximately 55–81% and increased temperature reduced larval survival by 48% as compared to the control.

This study reveals new, valuable information on how near shore stressors such as runoff and elevated temperature affect the early life stages of a Hawaiian coral. Results that increase the understanding of the impact of local stressors on early life stages can provide managers with sound science to develop management strategies for the conservation and protection of coral reefs. Managers will be able to use this information in coral reef management programs such as outplanting coral fragments. Outplanting of reef building corals into areas that have lost cover or onto artificial reefs has become a popular method of reef restoration. Growth of adult colonies and asexual reproduction through fission or fragmentation can increase coral cover, but the long-term success of these outplanted populations depends on the genetic diversity and successful recruitment of sexually produced offspring. Understanding how the early life stages of corals are affected by near shore stressors will assist managers with outplanting corals in suitable habitat for adult growth and reproduction as well as recruit new larval corals.

Future studies involving all of the near shore stressors should include varying concentrations of sperm, combinations of stressors, species with different reproductive strategies (brooding vs. spawning), and multiple coral species. It is important to use different concentrations of sperm because an optimal sperm concentration is (1) not realistic for *in situ* concentrations and (2) could mask deleterious effects of the near shore stressors. Also, availability of sperm has been shown to have a strong influence on successful fertilization in corals (*Ricardo, Clode & Humanes, 2015*). It has been shown that there is inherent variability in quality and sensitivity of coral gametes and their larvae from different nights of spawning and should be taken into consideration when conducting future experiments (*Hédouin & Gates, 2013*). It is also important to see whether the effects of near shore stressors change when combined. Therefore, a factorial design where near shore stressors are combined to determine if effects are additive, synergistic, or antagonistic (*Chui & Ang Jr, 2015*; *Humanes et al., 2017*). Lastly, it is important to study the effects of near shore stressors on different species of coral and corals with different reproductive strategies. Other studies have shown that different coral species from the same reef system respond differently to stressors (*Hedouin, Pilon & Puisay, 2015*; *Negri, Marshall & Heyward, 2007*).

It is important for the conservation and protection of coral reefs that effects of near shore stressors on early life stages of corals be studied. The resilience and recovery of coral reefs is highly dependent on successful reproduction and settlement of larval corals.

## ACKNOWLEDGEMENTS

Gin Carter, Julio Camperio Ciani, Alison Dygert, and Reuben Schleiger contributed to field work and gamete collection.

### Funding
The authors received no funding for this work.

### Competing Interests
The authors declare there are no competing interests.

### Author Contributions
- Claire V.A. Lager conceived and designed the experiments, performed the experiments, analyzed the data, prepared figures and/or tables, authored or reviewed drafts of the paper, and approved the final draft.
- Mary Hagedorn and Paul L. Jokiel conceived and designed the experiments, authored or reviewed drafts of the paper, and approved the final draft.
- Kuʻulei S. Rodgers conceived and designed the experiments, analyzed the data, authored or reviewed drafts of the paper, and approved the final draft.

### Field Study Permissions
The following information was supplied relating to field study approvals (i.e., approving body and any reference numbers):

This research was conducted under the Department of Land and Natural Resources-Division of Aquatic Resources Special Activity Permit No. SAP 2015-48.

## Data Availability

The raw measurements are available in the Supplementary File.

## Supplemental Information

Supplemental information for this article can be found online at http://dx.doi.org/10.7717/peerj.9415#supplemental-information.

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
