# Peer review of "The impact of short-term exposure to near shore stressors on the early life stages of the reef building coral Montipora capitata"

_PeerJ, doi:10.7717/peerj.9415_

## Round 0.1 · original submission · Major Revisions

Three expert reviewers have evaluated your manuscript and their comments can be seen below. As you can see the comments are favourable, but all reviewers have insightful and important comments and suggetions that need to be taken into consideration in a revision.

Reviewer 1 ·

Basic reporting

Relatively clearly written, the only ambiguity was regarding descriptions in the methodology (detailed later).

Coverage of the literature was OK but needs updating, particularly with respect to the effects of sediments on the early life stages of coral (detailed later)

Structure good, Figures and Table need more description, Raw data needs some extra details

Self-contained with relevant results and hypotheses – but some clarification needed to describe the effects on survival and settlement as being latent (not directly due to the treatments).

Experimental design

Original primary research within Aims and Scope of the journal - yes

Research question well defined, relevant & meaningful. It is stated how research fills an identified knowledge gap - more detail needed to show how this research fits with that of more recent studies.

Rigorous investigation performed to a high technical & ethical standard - yes

Methods described with sufficient detail & information to replicate - needs some clarification see general comments to the Author.

Validity of the findings

Results are valid but I would like to know if the methodology may have resulted in an experimental artefact that could affect the results. It is not clear to me exactly how this was done experimentally, but if the embryos were left with the original sperm for the first 24 h this could exacerbate the effect of all the treatments. Ideally the surviving successful embryos would be cleaned of sperm and placed in new seawater (or treatment seawater) and the experiment continued. Dying sperm will stick to sediments and increase bacterial loads, especially in high temperature treatments (maybe also low salinity treatments). The authors should clarify all of this in methods and discuss later if it is seen as an issue.

Additional comments

Abstract: needs considerable editing to make clear the sequence of exposures and the timing – the Abstract is far too ambiguous at present.

L32: “Effects of of sedimentation (100 mg l-1 and 200 mg l-1 ), osmotic stress through lowered salinity (28 ‰), and elevated temperature (31°C) on the successful fertilization, larval survival, and settlement” – were these separate, simultaneous or sequential? Please detail.

L32: The term “sedimentation” was used to describe two suspended (SS) concentrations. Sedimentaion means something different (the deposition of sediment), please change to “suspended sediment concentrations”

L35: I don’t think gametes can be exposed, for 24 h. After a few hours – 6? Gametes are either fertilised or dead? – rephrase please.

L36: How long were larvae exposed? How old were the larvae? Gametes do not develop into larvae until + 24 h. Please be more precise with descriptions of what exactly was exposed and for how long.

Introduction: Concise and clear but overlooks the most relevant prior literature

Line 45: “rapidly experiencing global decline” but you cite references 15 and 19 years old – please update

Line 55: more info on where “Kāne‘ohe Bay, O‘ahu” is in the world.

Line 69: how much warmer than the average summer maxima?

Line 75: update reference with more recent reviews (i.e. Jones et al., 2016 Mar. Pollut. Bull. 102:9-29)

Line 76: “However, very little is known about how these near shore stressors affect the early life stages of coral, especially in Hawaiian coral”. Maybe for Hawaiian coral but since 2015 there has been a very large body of work done on the effects of suspended sediments, sediment deposition and thermal stress on coral fertilisation, larval survival and larval settlement (see multiple relevant papers by Ricardo et al. and Humanes et al.,) and before that more than 10 studies on the effects of temperature alone on coral reproduction alone. Please refer to at least some of this very relevant literature.

Line 83: exposing gametes for 24 h. “gametes” are the eggs and sperm only. Please rephrase to let us know how old each of your life stages were.

Materials and methods: The treatment sequence is unclear and needs to be rewritten. It seems to me that the only treatment phase was during the first 24 h and then the embryos were moved back into control conditions? The larval survival and settlement should then be referred to as a latent effect from the original exposure during fertilisation and early embryo development.

L102: what was the temperature of the bay and the water in the aquaria? (bring some of the information from L136 forward to here”)

112: how big was the beaker, how long was it washed for? Why was is washed, “to remove….?”

L112: are 100 and 200 mg/L ecologically relevant? See Jones et al., 2016.
L136: what was the temperature range, how was it measured?

L136: was it the current in the tank that simulated “mile wave motion” – please explain so others can repeat
L152: what is the approximate sperm concentration. The work of Ricardo et al shows that sperm concentration has a very strong influence on the sensitivity of fertilisation to suspended sediments and is likely to also affect the sensitivity to thermal and osmotic stress.

Table 1. This table needs improving. It is hard to see how it represents five treatments.
I suggest making things unambiguous.
Each row should tell us what the conditions were in each of the five treatments
Control treatment: 0 mg/L SS 34 PSU 27oC
Medium SS: 100 mg/L SS, 34 PSU, 27oC
High SS: 200 mg/L SS, 34 PSU, 27oC
High temp: 0 mg/L SS, 34 PSU, 27oC
Low salinity: 0 mg/L SS, 28 PSU, 27oC

Line 156: if two sperm-egg bundle were in a single vial and this represents one cross. How is this single cross exposed to all five treatments? Was each cross divided into five separate vials? If so, what about the excess sperm?

L158: How many eggs (range or mean and SD?) were in each vial?

L158: here you need to tell us that the exposure for fertilisation was actually 8 hours

L163: Need to more clearly describe the exposure sequence (could do a diagram). Please let me know if I get this right:
1. Expose gametes to each of 5 treatments for for 8 h and fertilisation counted.
2. Total exposure of gametes and developing embryos was 24 h (there was sperm in these vials the whole time).
3. After 24 h the embryos were transferred to clean vials (how many embryos per vial?, was this also a cleaning step). Embryos now in clean seawater only = control conditions? Were they all kept at 27oC now or were the 31oC embryos kept at the elevated temperature?
4. Survival of early larvae were counted at T = 48 h (=24 h in the treatment conditions + 24 h in clean water at 27oC and normal salinity- this is not at all clear).
5. larvae transferred to 10 ml petri dishes, CCA added, water replaced every 2 d and settlement counted at 14 d (all in control conditions?).

L175: “metamorphosed” into what? Briefly define what this is with a reference.

Where is the statistics section in the M+M?

Results:

L191: were they statistically higher?

L199 – this is “latent” survival, after the initial 24 h exposure and 24 h at control conditions right?

L199: Please stat that the survival is a % of the successfully fertilised gametes. (not a % of the total eggs used?).

Figure 2. what is the number of replicate vials? Needs to be in the caption. Say that each step is % of the previous. For example, survival is a % of the successfully fertilised. Settled is a % of those that survived.

It would be useful to also include the total success to settlement relative to the number of eggs in each treatment.

Discussion: Missing comparisons with the most recent and relevant work on suspended sediments, needs more discussion on why these life stages are/are not affected.

Line 220: Stat that this is a one-off 24 h exposure and that survival and settlement were due to “latent” effects of the original treatment.

Lines 228-235: This section should instead focus on the work of Ricardo et al and Humanes et al. Ricardo outlined shortcomings of the early Gilmour work and used measured sediment concentrations and tested the influence of sperm concentration on the fertilisation success. Humanes et al performed similar tests to those here (but included sediment x temperature treatments also).
It is critical that you contrast your work with that of others more specifically. You did not directly test the effects of sediments on larval settlement. The exposure was for the first 24 h so the effects on survival and settlement at latent effects (this is legitimate but you need to emphasise it). Recent work has shown that coral larvae are can be very resistant to suspended solids (mechanism were described) but settlement is very sensitive to fine layers of sediments. There should be discussion about the likely sensitivity of the test given the sperm concentrations used and the sediment type used (Ricardo extensively tested the effects of sediment type on fertilisation inhibition – type has a major effect.

Line 239 – Ricardo et al also demonstrated effects of films of sediment on larval settlement.

Line 246: what are the likely mechanisms for effect of low salinity on embryo and larval health?

Discuss future work with combinations of the stressors with Hawaiian corals (Similar to Humanes work).

Raw Data
Columns A and C do not have labels – what are they?

Reviewer 2 ·

Basic reporting

This paper studied the effects of temperature, salinity and sediment on fertilization, larval survival and settlement of the reef building coral M. capitata.
The authors performed a very simple experiment where they exposed eggs and embryos to different stressors independently.
It was very interesting to see the drastic effects of low salinity in larval survival and how fertilization and settlement were less affected by different stressors. I think this type of experiments are really important to better predict the resilience of coral reef ecosystems and the stressor effects on future generations.

I wish the authors had done simultaneous exposure of stressors, it would have been a more realistic and novel experiment, but I also understand the difficulty of performing this type of experiments.

The manuscript overall is not well written, the methods lack detail and both the introduction and discussion lack depth.
What are the hypotheses?
How could the stressors interact in the field?
Is there any evidence of synergistic or antagonistic effects?
Why do you think larvae was the more vulnerable stage?
What cellular/molecular mechanisms could be responsible for the patterns observed?
What is the ecological relevance of your results?
These questions need to be addressed in introduction and/or discussion.

ABSTRACT
Line 29: “It is critical…” This sentence is not well connected with the previous one.
Line 33: remove “osmotic stress through”
Line 33: add space “31 C”
Line 35: confusing sentence, you need to clarify that that eggs were not exposed to stressors simultaneously.
Line 38-41: I found this last sentence very hard to understand, I suggest authors to rewrite the end of their abstract. Why would you say that sediment stress is ongoing whereas “rain stress” is episodic. What do you mean by sedimentation? In many cases, sedimentation is associated with rain events. You talk about currents and transport, but you have no evidence on this. What do you mean by “improved” conditions.
Keep in mind that larvae can also move and change their location vertically as well.

INTRODUCTION
Line 56: Paragraph lacks order. I suggest you delete “ …and recently consecutive bleaching….(Bahr et al. 2015b).” Focus first on the freshwater, then talk about temperature stress/bleaching.
Lines 63-67 repetitive, please condense.
Line 71: (Bahr…): You can not start the sentence with parenthesis.
Line 72: Weird/incomplete sentence, you first talk about recovery and then mortality. What did Bahr et al. 2017 found in the context of recovery.
Line 75: Be specific “… on adult coral PERFORMANCE”
Lines 80-82: repetitive
Line 83: clarify that is independent exposure to stressors
Line 86: Please include ranges of environmental conditions in Hawaii (relevant to the stressors in your experiments).
Line 89: What do you mean by “maintenance”? Change word choice.

Experimental design

METHODS
WHAT IS YOUR SAMPLE SIZE AND REPLICATION?
Line 95: Location, coordinates?
Lines 101, 102: corals (plural)
Lines 108-115: The methods for the sediment treatment are very obscure.
Why was the supernatant decanted?
Why <63um?
You need to include more detail in how you performed your weight to dry weight ratio.
Line 122-123: Sentence not well written and hard to understand.
Line 132: Add space “27 C” and “31 C”.
Lines 135-136: Not clear how you achieved “wave motion/agitation“ inside the vial. Floating rack?
Line 144-145: What do you mean by “pooled”? Be specific here (between individuals, high egg concentration?).
Line 145: Why “pooling” can affect the egg membrane?
Line 147: Replace Hagedorn et al. 2015 with Mate et al. 1997.
Line 153-56: Unclear if exposure was simultaneous or independent, please clarify.
Also, table 1 and its legend were confusing.
Line 212: confusing

Validity of the findings

DISCUSSION
Line 239: Replace “production” with “development”
Line 240: Not sure what you mean with “covered in varying levels”, be clear and specific here.
Line 246: Effects of LOW salinity….(include “low”)
Line 260-261: what do you mean by “well-mixed oceanic systems?”
Line 275: “…control and the treatments.” Do you mean temp, treatments? Be specific.
Line 276: Low replication? Include sample size
Line 283: what do you mean by “large scale” storms?
Line 285: what “magnitude”?
Line 285: … that MAY result in…
Line 284-287: Several ideas and not well connected.
Line 287: Replace “treatments” with “stressors”…
Line 293-295: What type of management strategies? In what context can your results inform conservation strategies? What type of mitigation efforts? What is the association with your data and predictive modeling?

Reviewer 3 ·

Basic reporting

I have structured my feedback per the journal's instructions to reviewers and not in order of importance.


1. English language
There are frequent typos, grammatical errors, and vague phrases in the text that should be addressed. I have made note of some here:
L26 - the use of "maintenance" is unusual. Is there a more appropriate word choice?
L27 - "due to greenhouse gases" implies a more direct relationship than in reality, implying that there are greenhouse gases in reef waters that are causing them to warm
L29 - missing word
L44 - insert comma after world
L52 - "all of these" is vague
L74 - insert comma after local. Also, elevated temperature is not a local, near-shore stressor. It is a global stressor.
L100 - typo
L151 - "this method" is vague
L172 - "and covered" - what does that refer to - the dish? Here and in the next sentence, make sure the subject of the sentence makes sense with both clauses.
L201 - ditto

2. Background and context
The justification of the treatment levels and the exposure length needs more support. How do you know that the levels are ecologically relevant? Why is this exposure length relevant? Include evidence that demonstrates that Kaneohe Bay experiences these treatment levels. The fact that the Bay has experienced warmer summers in 2014 and 2015 is not justification that the bay is warming due to climate change.

3. Article structure, figures, tables, data
The article is structured well with appropriate sections. Figure 1 and Table 1 do not add substantial value to the manuscript, and I recommend that they are removed because the information is sufficiently explained in the text. The authors should include a table with all of the statistical results, which could be included as supplementary information. I recommend breaking Figure 2 into three figures that better reflect the statistical analyses and comparisons discussed (i.e. sedimentation, salinity, temperature).

4. Results are relevant to hypotheses

Experimental design

I have structured my feedback per the journal's instructions to reviewers and not in order of importance.

1. Research question and knowledge gap
I found the introduction to be misleading about the knowledge gap. For example, L74-78 states that the effects of sedimentation, low salinity, and warming have been characterized for adult corals but not for early life stages. However, in the discussion section, many studies looking at effects of these stressors on coral fertilization, larval survival, etc are referenced. The authors need to more clearly define the knowledge gap that this study fills.

Why were the stressors not examined in combination? I imagine that sedimentation, low salinity, and elevated temperature often co-occur on these reefs. Their combined effects may then be quite relevant to examine.

The research question is not clearly defined

2. Rigor of investigation
The level of replication needs to be clearly described. Were there replicate vials per parental cross per treatment condition? How many crosses or parental representation were present on each day?
Further, the experiment was conducted for 34 crosses by using egg-sperm bundles released on different days. The crosses need to be included in the statistical framework. Also, previous work with coral larvae has shown that offspring released on different days from the same parents have different physiological characteristics and sensitivities. The potential role of this phenomenon in the outcome of the present study should be acknowledged and discussed.

3. Methods were not described in sufficient detail for replication (See number 2 above)

Validity of the findings

I am unsure of the validity of the findings without knowing about the replication as described above.

The discussion should be revised to make sure that all of the results are interpreted (fertilization is over-represented) and to include a broad sense of how these results will improve our understanding of coral reef ecology under climate change.

Additional comments

The authors conducted a study to look at effects of sedimentation, low salinity, and warming on early life stages of a coral. These results of the study provide relevant information regarding the challenges that near-shore reefs face. The article requires significant revisions, most importantly a clear framing of the knowledge gap and more information about experimental design, and I hope the feedback is helpful in efforts to improve the manuscript.

---

## Round 0.2 · Minor Revisions

Two expert reviewers have re-evaluted your manuscript. One of the reviewers has made some good suggestions to improve the manuscript.

Reviewer 1 ·

Basic reporting

Much improved following revision. Clearly written, appropriately referenced, good tables and figures and self-contained from hypothesis to conclusions

Experimental design

Clarified in this second version. Original, a meaningful research question, scientifically rigorous, and methods now well described.

Validity of the findings

Scientifically valid and well described

Additional comments

The authors have comprehensively addressed all my previous comments. This is a nice paper and ready for publication. Well done.

Reviewer 2 ·

Basic reporting

I sincerely apologize for my delay with this review.
I have now reviewed the updated manuscript and have seen the author’s responses to the reviewers’s comments. The authors have a done a good job addressing most comments and the manuscript in in much better shape.
However, there are still several grammatical errors that make some sentences confusing and/or do not accurate describe the methodology.

I have included revisions on the word document (highlighted in yellow and green).

I am also including additional comments.

Lines 160-164: The “compensation” part does not make a lot of sense. I suggest you delete these sentences and just add in the discussion the fact that it is important to take in consideration that sensitivity may vary depending on the timing of the release (Hedouin & Gates 2013).

Lines 229-232: Very confusing sentence, check grammar

Line 241: Unclear how did you obtain the value of 5 x 105 cells/mL
The sperm concentration of a bundle should not be “standardized” by volume. Because volume depends on the dilution you are working with, so is irrelevant. Please check grammar and correct word choice to have accurate meaning.

Line 244-245: “Each cross had 6 scintillation vials…”
This does not make sense, please change word choice to accurately reflect what you did.
Ex: each cross was exposed

Line 353-354: Very confusing sentence, since you did not find effects of treatments on fertilization.

After these revisions (and in word document), the manuscript should be ready to be accepted.

Kind regards,

Experimental design

ok

Validity of the findings

ok

Annotated reviews are not available for download in order to protect the identity of reviewers who chose to remain anonymous.

---

## Round 0.3 · accepted · Accept

I am satisfied with the changes made to the manuscirpt. Minor errors should be corrected at proofs stage eg line 39: These data shows should read These data show (because the word data is plural).